# An Updated Review of the Invasive *Aedes albopictus* in the Americas; Geographical Distribution, Host Feeding Patterns, Arbovirus Infection, and the Potential for Vertical Transmission of Dengue Virus

**DOI:** 10.3390/insects12110967

**Published:** 2021-10-26

**Authors:** Julian E. Garcia-Rejon, Juan-Carlos Navarro, Nohemi Cigarroa-Toledo, Carlos M. Baak-Baak

**Affiliations:** 1Centro de Investigaciones Regionales, Laboratorio de Arbovirologia, Universidad Autonoma de Yucatan, Merida 97069, Yucatan, Mexico; julian.garcia@correo.uady.mx; 2Grupo de Investigación en Enfermedades Emergentes, Desatendidas, Ecopidemiología y Biodiversidad, Facultad de Ciencias de la Salud, Universidad Internacional SEK, Quito 170107, Ecuador; 3Centro de Investigaciones Regionales, Laboratorio de Biología Celular, Universidad Autonoma de Yucatan, Merida 97069, Yucatan, Mexico; nohemi.cigarroa@correo.uady.mx

**Keywords:** Asian tiger mosquito, feeding pattern, minimum infection rate, emerging arboviruses, Dengue virus

## Abstract

**Simple Summary:**

Currently, the Asian tiger mosquito Aedes albopictus Skuse is present on all continents except Antarctica. Efficiency as a vector of Ae. albopictus is different by geographic region. In areas where Aedes aegypti is absent, the Asian mosquito is the main vector of arboviruses such as dengue, Zika, and chikungunya. In the Americas, Ae. albopictus occupies the same ecological niches as Ae. aegypti. It is difficult to incriminate the Asian mosquito as the cause of autochthonous arbovirus outbreaks. However, evidence suggests that Ae. albopictus is very effective in transmitting endemic arboviruses (such as dengue) both horizontal and vertical transmission. Aedes albopictus could be useful as a sentinel species to monitor dengue virus in interepidemic periods.

**Abstract:**

*Aedes* (*Stegomyia*) *albopictus* is a mosquito native to Southeast Asia. Currently, it has a wide distribution in America, where natural infection with arboviruses of medical and veterinary importance has been reported. In spite of their importance in the transmission of endemic arbovirus, the basic information of parameters affecting their vectorial capacity is poorly investigated. The aim of the work was to update the distribution range of *Ae. albopictus* in the Americas, review the blood-feeding patterns, and compare the minimum infection rate (MIR) of the Dengue virus (DENV) between studies of vertical and horizontal transmission. The current distribution of *Ae. albopictus* encompasses 21 countries in the Americas. An extensive review has been conducted for the blood-feeding patterns of *Ae. albopictus*. The results suggest that the mosquito is capable of feeding on 16 species of mammals and five species of avian. Humans, dogs, and rats are the most common hosts. Eight arboviruses with the potential to infect humans and animals have been isolated in *Ae. albopictus*. In the United States of America (USA), Eastern equine encephalitis virus, Keystone virus, La Crosse Virus, West Nile virus, and Cache Valley virus were isolated in the Asian mosquito. In Brazil, Mexico, Colombia, and Costa Rica, DENV (all serotypes) has been frequently identified in field-caught *Ae. albopictus*. Overall, the estimated MIR in *Ae. albopictus* infected with DENV is similar between horizontal (10.95) and vertical transmission (8.28). However, in vertical transmission, there is a difference in the MIR values if the DENV is identified from larvae or adults (males and females emerged from a collection of eggs or larvae). MIR estimated from larvae is 14.04 and MIR estimated in adults is 4.04. In conclusion, it has to be highlighted that *Ae. albopictus* is an invasive mosquito with wide phenotypic plasticity to adapt to broad and new areas, it is highly efficient to transmit the DENV horizontally and vertically, it can participate in the inter-endemic transmission of the dengue disease, and it can spread zoonotic arboviruses across forest and urban settings.

## 1. Introduction

*Aedes* (*Stegomyia*) *albopictus* Skuse is a mosquito native to Southeast Asia, colloquially known as the Asian tiger mosquito or Asian mosquito. The mosquito was described by Skuse (1894) in the city of Calcutta, India [1,2]. At the beginning of 2000′s, its importance as a vector of arboviruses was restricted to Asian and African countries [1]. Currently, *Ae. albopictus* is present on all continents except Antarctica [3]. It has been observed that once established in new geographic areas, it can become involved in the natural cycles of arbovirus transmission. For example, in Europe it has colonized several countries and was involved in dengue outbreaks in France, Italy, and Spain [4,5,6]. In Italy, the genome of the chikungunya virus was identified in *Ae. albopictus* and it was incriminated as the vector that caused the local outbreaks of chikungunya fever [7]. Likewise, autochthonous cases of Zika fever occurred in France and *Ae. albopictus* was suspected as the transmitter of the virus [8]. In America, dengue virus is the most important mosquito-borne viruses in terms of its global impact on human morbidity and mortality. Approximately 23 million dengue cases were registered across the Americas between 1980 and 2017. In 2019 there was a resurgence of the dengue virus, reaching 3.1 million cases throughout the region [9]. chikungunya and Zika viruses are emerging viruses in America that have caused explosive outbreaks from 2013 to 2016, which has since subsided [10]. *Aedes aegypti* is the main vector of dengue, Zika, and chikungunya viruses in the region [10,11,12,13,14]. The antecedents demonstrate that *Ae. albopictus* can transmit these viruses [5,7,11,14]; therefore, it is considered a species with the potential to increase the risk of arbovirus transmission in America.

*Aedes albopictus* has a wide distribution in America [3,15,16,17,18,19,20,21,22,23,24,25,26,27,28,29,30,31,32,33,34,35]. The USA [1] began the surveillance of the geographic distribution throughout America since 1985, when it became known that the Asian mosquito had colonized the state of Texas, and studies were carried out on the pattern of blood-feeding and the transmission of arboviruses. Studies on the blood-feeding pattern of *Ae. albopictus* have been carried out in the United States of America and Brazil [36,37,38,39,40,41,42,43,44,45,46]. The results show that it is an opportunistic mosquito. DNA from humans and a diverse range of wild and domestic animals have been identified in the blood meal of the mosquito [38,39,41,45].

In America, arboviruses of medical and veterinary importance have been isolated in field-caught *Ae. albopictus* [47,48,49,50,51,52,53,54,55,56,57,58,59,60,61,62,63,64,65,66,67,68]. Notably, the Asian mosquito has a great capacity to acquire arboviruses and transmit them to its offspring. The findings of transovarian transmission have been consistent and very frequent [56,57,58]. Studies carried out in North and South America have found the dengue (all serotypes), Zika and La Crosse viruses in larvae and males of *Ae. albopictus* [57,60,62,63,64]. Evidences suggest that the mosquito may have a reservoir role for the dengue virus by keeping it silent in nature [56]. In Brazil, the detection of DENV-3 in males of *Ae. albopictus* was carried out in years in which no autochthonous human cases with this serotype were recorded, suggesting that the silent circulation of DENV-3 occurs by a vertical transmission mechanism [68]. Additionally, yellow fever virus was isolated in *Ae. albopictus* females in Rio de Janeiro State, which could imply that it could be acting as an additional jungle or rural vector causing a possible transmission bridge to the urban area [47]. 

At this point, the distribution range of *Ae. albopictus* in the Americas was updated, the blood feeding patterns were reviewed, and the minimum infection rate of the dengue virus between studies of vertical and horizontal transmission was compared.

## 2. Selection Criteria and Search Strategy

The analysis only included works carried out in the Americas (North, South, Central, and the Caribbean), with topics focused on the first report of *Ae. albopictus* from each American country, blood feeding patterns, and reports of natural infection with arbovirus. 

Databases of Google Scholar, PubMed Health (National Center for Biotechnology Information at the National Library of Medicine), SciELO (Scientific Electronic Library Online), and Web of Science (Thompson Reuters) were used for the literature review. The search was done with combination of keywords including “*Aedes albopictus*” AND “first report”, “first record”, “new records”, “blood meal”, “feeding pattern”, “arbovirus” “Dengue”, “Zika”, “chikungunya”, “America”. Additional references were facilitated by colleagues. 

Importation of references and removal of duplicate references were done by using the bibliographical software package, Mendeley version 1.19.8 (Elsevier, Amsterdam, Netherlands). All titles, abstracts and selected full reports were screened independently by two authors based on the inclusion and exclusion criteria. Discrepancies were resolved by consensus.

### Statistical Analysis

Test for the difference of proportions was used to compare the positive pools between studies with vertical and horizontal transmission cycle. The MIR values of each study were extracted manually and were organized in an Excel sheet. When the work did not include the MIR, it was calculated with the following formula: (1)$$\mathrm{MIR}=\frac{\mathrm{Number}\;\mathrm{of}\;\mathrm{positive}\;\mathrm{pools}}{\mathrm{Total}\;\mathrm{numbers}\;\mathrm{of}\;\mathrm{mosquitoes}\;\mathrm{tested}}\times 1000$$

Host frequencies identified in blood meals of *Ae. albopictus* were extracted from each work and were organized in an Excel sheet. Statistical analyses were performed by using *R* statistical programming language version 4.0.2, and results were considered statistically significant when *p* ≤ 0.05.

## 3. Chronological Order of the First Reports of *Ae. albopictus* in the Americas

The current distribution of *Ae. albopictus* encompasses 21 of 44 countries in the Americas, although the colonization pattern is different in each country (Table 1) and Chile and Peru have not reported any data yet. Previously, Kramer and collaborators [3] conducted a global compendium of the distribution of *Ae. albopictus* and described its presence in 16 countries of the Americas. According to reports, the mosquito has presented an erratic distribution, but with great rapidity in its movement through America. The introduction of *Ae. albopictus* in America was divided into four periods. In the first period (1983–1990), the Asian mosquito was reported in three countries. The first report occurred in the USA in 1983, when a single adult of *Ae. albopictus* was captured in a cemetery in Memphis, Tennessee [15]. Three years later, five male and six female mosquitoes with similar characteristics to the Asian mosquito were captured and their identity was confirmed as *Ae. albopictus* in Brazil (1986) [16]. In Mexico, the Asian mosquito was reported for the first time in 1988: the larvae were collected in tires [17]. In the second period (1993–1998), the Asian mosquito was reported in six countries including the Dominican Republic, Cuba, Guatemala, the Cayman Islands, Colombia, and Argentina [18,19,20,21,22,23]. Reiter [15] mentions that *Ae. albopictus* was reported in Bolivia and El Salvador, but there are no reports that confirm this. Their presence in these countries is not currently recognized. In the third period (2000–2010), the mosquito significantly expanded its distribution to ten countries, including Bermuda, Canada, Trinidad and Tobago, Panama, Uruguay, Nicaragua, Costa Rica, Venezuela, Belize, and Haiti [24,25,26,27,28,29,30,31,32,33]. In the fourth period (2011–2021), the presence of the mosquito was only reported in Ecuador in 2017 and in Jamaica in 2018 [34,35]. It is well documented that the introduction of *Ae. albopictus* into America occurred through tires and bamboo stumps imported from Japan. It is also hypothesized that the massive distribution of the mosquito occurred through the export of used tires among countries in the Americas, Europe and Asia [1,15,16,17]. Within countries, automobiles are believed to contribute to the distribution [69].

## 4. Blood Feeding Pattern of *Ae. albopictus*

There are a total of 11 published papers on the blood feeding pattern of *Ae. albopictus*; nine of them were carried out in the USA and two in Brazil. The first four studies used the serological precipitin test and ELISAs to identify the identity of the vertebrate hosts [36,37,38,39]. Seven publications used PCR to identify host DNA [40,41,42,43,44,45,46]. Moreover, 1925 individual mosquitos were tested in total. In 85.56% (1647/1925) of the mosquitos, the host was identified at the species level, which comprised 16 species of mammals and five species of avian (Table 2). Despite the ability of *Ae. albopictus* to feed on the blood of different vertebrate taxa, 98.70% (1900/1925) corresponded to mammals. The human (*Homo sapiens*), the domestic dog (*Canis lupus*), the brown rat (*Rattus norvegicus*), and the domestic cat (*Felis silvestris*) are the most frequent hosts in the publications and with more specimens analyzed (Table 2). 

The frequency of blood feeding of *Ae. albopictus* on a particular host determines the risk of pathogen transmission. According to studies mainly published in the USA, the Asian mosquito has an anthropophilic tendency, although in the absence of humans it can feed on 15 other species of mammals and five species of avian. The method and the place of capture of *Ae. albopictus* was decisive to identify DNA of hosts in the blood meals of the mosquito. Most females of *Ae. albopictus* with human blood were captured with the human bait method and aspirated from mosquitoes indoors and outdoors [37,39,41,43]. The other works captured *Ae. albopictus* in the forest or habitats with abundant vegetation. For this reason, the number of wild species in the blood meals of the Asian mosquito was very diverse [36,38,40,42,44,45,46]. In the USA, the feeding frequency of *Ae. albopictus* on avian and wild mammals partly explains the isolation of zoonotic arboviruses (Table 3) [48,49,50,51,62].

## 5. Natural Infections of *Ae. albopictus* with Arboviruses

In the Americas, there are 24 published papers on the identification of arboviruses in field-caught *Ae. albopictus* with the potential to infect humans and animals [11,14,47,48,49,50,51,52,53,54,55,56,57,58,59,60,61,62,63,64,65,66,67,68]. Ten of the findings were obtained in Brazil, six in the USA, four in Mexico, three in Colombia, and one in Costa Rica (Table 3). Eastern equine encephalitis virus, Keystone virus, Cache Valley virus, La Crosse virus, West Nile virus, dengue virus (all serotypes), yellow fever virus, and Zika virus were the arboviruses identified (Table 3). 

Notably, 66.66% (16/24) of the publications reported the genome of the dengue virus in the Asian mosquito, although the presence was confirmed in only four studies via viral isolation [57,58,61,66]. In decreasing order, the most frequent serotypes in the publications are DENV-2 (*n* = 8), DENV-3 (*n* = 5), DENV-1 (*n* = 4), and DENV-4 (*n* = 3). On the other hand, the Zika virus was identified in *Ae. albopictus* in six studies carried out in Brazil (*n* = 4) and Mexico (*n* = 2) [14,47,53,54,57,59]. 

*Aedes albopictus* has a wide distribution in America. Despite this fact, the natural infection of *Ae. albopictus* with arboviruses of medical and veterinary importance has been reported only in five countries. Currently, eight arboviruses have been isolated in field-caught *Ae. albopictus* (Table 3). The Asian mosquito is a competent experimental vector of 16 arboviruses that circulate in America. Among the arboviruses experimentally transmitted by *Ae. albopictus,* there are viruses of the families Flaviviridae (dengue virus, Zika virus, West Nile virus, and yellow fever virus), Togaviridae (chikungunya virus, Eastern equine encephalitis virus, Mayaro virus, Ross River virus, Sindbis virus, Western equine encephalitis virus, and Venezuelan equine encephalitis virus), and Peribunyaviridae (Jamestown Canyon virus, Keystone virus, La Crosse virus, Potosi virus, and Rift Valley virus) [70,71,72]. 

In 2013, the chikungunya virus (CHIKV) emerged in the Americas and caused local outbreaks of chikungunya fever. To date, no natural infection with this virus has been reported in *Ae. albopictus* [12,13]. The Asian mosquito is an efficient vector of the epidemic mutant strain CHIKV_0621 of the East–Central–South African (ECSA) genotype [73], which, caused autochthonous cases of CHIKV in Indian Ocean [74]. Today, the circulation of the mutant strain in America is not reported.

The first findings of dengue virus in the Asian mosquito were obtained through vertical transmission. In Brazil, DENV-1 was isolated from two pools of mosquito larvae in 1993 [58]. Two years later, during a dengue outbreak in Mexico, DENV-2 and DENV-3 were isolated from a pool of 10 males of *Ae. albopictus* [61]. Again, in Brazil DENV-3 was identified in three larval pools in 1999 [63].

The Eastern equine encephalitis and Keystone were the first arboviruses isolated from the Asian mosquito, which were captured in a tire dump in Florida [48]. This was the first evidence of *Ae. albopictus* as a potential arbovirus vector in the region. In the USA, Eastern equine encephalitis virus, Keystone virus, La Crosse Virus, West Nile virus, and Cache Valley virus were isolated in field-caught *Ae. albopictus* [10,11,12,14,15,27]. Most of these viruses were identified by horizontal transmission, except for the La Crosse virus (LCV), which was isolated in two pools of females emerged from larvae collected in homes of patients with confirmed LAV encephalitis. 

More than 70% of the publications of *Ae. albopictus* naturally infected with the dengue and Zika viruses come from Brazil, Mexico, Colombia, and Costa Rica, which are dengue-endemic countries and between 2014 and 2018 there was active transmission of the Zika virus [10,11,14,47,52,53,54,55,57,58,59,60,61,63,64,65,66,67,68]. Notably, in 9 out of 10 studies carried out in Brazil, dengue (all serotypes), Zika, and yellow fever viruses were transmitted through a transovarial route [47,57,58,59,63,64,65,66,68]. Future studies should focus on finding out if there is an evolutionary relationship of arbovirus adaptation with vertical transmission of *Ae. albopictus*.

## 6. The Minimum Infection Rate of the Dengue virus

Some authors have pointed out that the MIR underestimates viral infection since it assumes that a positive pool corresponds to a single infected mosquito [60]. To correct the bias, some researchers pool a small number of mosquitoes (≤10). Despite the bias, it is a parameter that is still used to find the probability of infected mosquitoes [55,60]. It is used as a measure to determine the capacity of *Ae. albopictus* as an efficient vector of arbovirus whose studies reported 12 vertical and 10 horizontal transmissions (Table 3]. Two studies carried out in Costa Rica and Colombia reported both types of transmission [55,60]. In 14 publications on vertical transmission [47,55,57,58,59,60,61,62,63,64,65,66,67,68], 19,435 mosquitoes organized in 792 pools were tested. Overall, 8.45% (67/792) of the pools were positive. The overall MIR was 3.45 per 1000 mosquitoes tested.

In contrast, in 12 publications on horizontal transmission [11,14,46,48,49,50,51,52,53,54,55,60], 53,566 mosquitoes organized in 5,956 pools were tested. Overall, 0.97% (58/5,956) of the pools were positive. The overall MIR was 1.08 per 1000 mosquitoes tested. The higher percentage of positive pools in vertical transmission contributed to a statistically significant difference compared to horizontal transmission (*X*^2^ = 215.46, d.f = 1, *p* ≤ 0.001).

When only DENV in vertical transmission was analyzed (Figure 1A,B), 57 positive pools of 6,883 mosquitoes were found [55,57,58,60,61,63,64,65,66,67,68]. Meanwhile, in horizontal transmission, 17 positive pools of 1,552 mosquitoes were found [11,52,54,55,60]. Therefore, MIR was slightly high in horizontal transmission (10.95) compared to vertical transmission (8.28) (Figure 1C,D). The estimated MIR in females of *Ae. albopictus* infected with DENV is in the range of 5.95 to 43.85 [11,52,54,55,60]. These values are similar to those estimated for *Ae. aegypti* [11,60]. This fact suggests that the Asian mosquito is also potentially effective in transmitting the DENV. The difference in the effectiveness to transmit the DENV is probably due to the endophilic and anthropophilic behavior of both mosquitoes. *Aedes aegypti* feeds almost exclusively on humans and rest inside the homes, taking more than one blood meal in each gonotrophic cycle [75,76,77]. While *Ae. albopictus*, although it feeds on humans, it is more opportunistic in its diet and prefers forest environments or areas with a lot of vegetation [36,37,38,39]. 

In a deeper analysis, it has been observed that there is a difference in the MIR values in vertical transmission if the DENV is identified from larvae [58,63,64,65] or adults emerged from a collection of eggs or larvae [55,57,60,61,66,67,68]. MIR estimated from larvae is 14.04 (41/2921) and 4.04 (16/3962) in adults per 1000 mosquitoes tested. The estimated MIR in *Ae. albopictus* larvae infected with DENV is in the range of 1.77 to 28.20 [58,63,64,65]. Pessanha and collaborators [65] estimated the MIR of 138 in *Ae. aegypti* larvae infected with DENV. The identification of DENV from the larvae of both mosquitoes is apparently very successful. 

In many geographic areas of the Americas, *Ae. albopictus* occupies the same ecological niches as *Ae. aegypti*. It is difficult to incriminate the tiger mosquito as the cause of autochthonous arbovirus outbreaks [11,60,61]. In horizontal transmission, *Ae. aegypti* is considered the main vector of dengue, Zika, and chikungunya viruses in American countries [11,12,13,14,53], while the Asian mosquito is considered a secondary vector in the transmission of these viruses. However, evidence suggests that *Ae. albopictus* is effective in transmitting the dengue and Zika both horizontal and vertical transmission [11,14,47,52,53,54,55,57,58,59,60,61,63,64,65,66,67,68]. Notably, 11 out of 14 publications refer to transovarial transmission of the dengue virus [55,57,58,60,61,63,64,65,66,67,68]. This has several aspects; the dengue virus can remain and persist silently during interepidemic periods [56,68]. The dispersal of eggs and larvae of *Ae. albopictus* infected with dengue and Zika viruses can cause the emergence and re-emergence of arboviruses and modify the local epidemiological pattern [47,58,59,62,63,64,65,68]. Transovarial transmission ensures the presence of arboviruses in *Ae. albopictus* regardless of blood feeding on viremic hosts. The occurrence of male mosquitoes infected by transovarial transmission suggests an equal probability of infection of the females of the same batch. Females of *Ae. albopictus* would not have to go through the extrinsic incubation period to transmit the virus to humans, which would enhance the dynamics of dengue transmission [60]. In addition, serotypes and genotypes not associated with autochthonous outbreaks have been detected during transovarial transmission. In Brazil, genotype III of DENV-3 was detected in larvae of *Ae. albopictus* collected in 1999 [63]. Nevertheless, DENV-3 (genotype III) was first isolated as an autochthonous case from a 40-year-old woman residing in Sao Paulo, Brazil [78], which suggests that this serotype was present in Brazil one year before its detection. Similarly, DENV-3 was detected in males of *Ae. albopictus* in years when no human autochthonous cases of this serotype were recorded in São Paulo, Brazil [68].

## 7. Concluding Remarks and Future Prospects

Despite the importance of *Ae. albopictus* as a vector and reservoir of dengue virus, few studies have evaluated the vectorial capacity in the Americas. Studies should focus on gonotrophic cycle length, dispersion range, daily survival probability, parity index and the proportion of bites made by females on humans. Likewise, in Asian mosquito populations, the susceptibility status and genes associated with resistance to insecticides used by local health services should be monitored. Finally, it is important to highlight that *Ae. albopictus* is an invasive mosquito with wide phenotypic plasticity to adapt to broad and new areas, it is highly efficient to transmit the DENV horizontally and vertically, it can participate in the inter-endemic transmission of the dengue disease, and it can spread zoonotic arboviruses across urban and peri-urban settings as well as natural settings. According to the MIR values of DENV, which were similar in horizontal (MIR = 10.95) and vertical transmission (MIR = 8.28), *Ae. albopictus* could participate in the natural cycle of transmission of DENV horizontally as the main vector *Ae. aegypti* and could also be useful as a sentinel species to monitor DENV in inter-epidemic periods.

## Figures and Tables

**Figure 1 insects-12-00967-f001:**
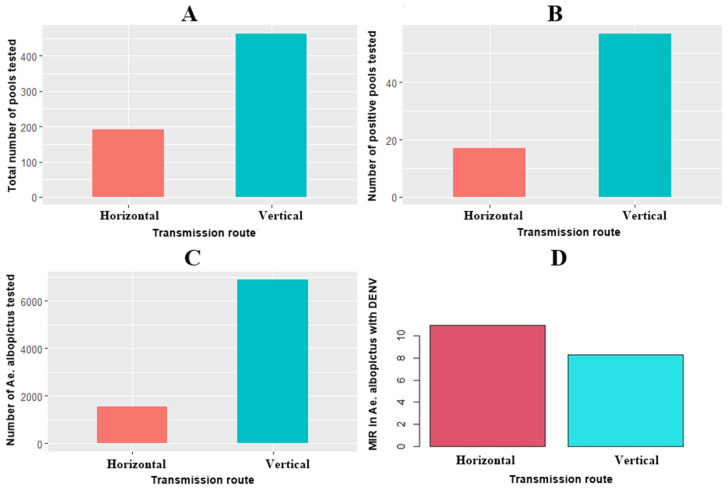
*Aedes albopictus* infected with DENV. (**A**) Total number of pools tested, (**B**) Number of positive pools tested, (**C**) Number of *Ae. albopictus* tested, and (**D**) MIR.

**Table 1 insects-12-00967-t001:** Chronological summary of publications on the first reports of *Ae. albopictus* in American countries.

Year of the First Report	Country	Collected Stage of the Mosquito	Author
1983	USA	A single adult collected	[15]
1986	Brazil	Captured five males and six females	[16]
1988	Mexico	Larvae collected in tires	[17]
1995	Cuba	Larvae collected	[18]
1993	Dominican Republic	Larvae collected in tires	[19]
1995	Guatemala	Larvae collected in tires, glass bottles, and metal drums.	[20]
1997	Cayman island	Larvae collected	[21]
1998	Colombia	Captured adults	[22]
1998	Argentina	Larvae and pupae collected	[23]
2000	Bermuda Island	Larvae collected	[24]
2002	Panama	Larvae collected	[25]
2001	Canada	Two adults captured	[26]
2002	Trinidad and Tobago	Eggs collected with ovitrap	[27]
2003	Uruguay	Adults captured	[28]
2003	Nicaragua	Larvae collected	[29]
2007	Costa Rica	Larvae collected	[30]
2009	Venezuela	Larvae collected	[31]
2009	Belize	Adults captured	[32]
2010	Haiti	Larvae collected	[33]
2017	Ecuador	Captured 5 males and 16 females	[34]
2018	Jamaica	Six females captured	[35]

**Table 2 insects-12-00967-t002:** DNA of vertebrate hosts identified in blood meals of *Aedes albopictus.*

Family	Vertebrate Host	Total Identified	Author
	Mammalian		
Hominidae	*Homo sapiens*	579	[36,37,38,39,41,42,43,44,45,46]
Muridae	*Rattus norvegicus*	227	[37,38,39,44,45,46]
Felidae	*Felis silvestris*	252	[38,39,41,43,44]
Canidae	*Canis lupus*	217	[36,38,39,40,41,43,44,45,46]
Sciuridae	*Sciurus carolinensis*	110	[41,43]
Leporidae	*Sylvilagus floridanus*	95	[37,38,43]
Cervidae	*Odocoileus virginianus*	52	[36,38,41,44]
Equidae	*Equus caballus*	49	[39,40,41]
Bovidae	*Bos taurus*	29	[36,38,39]
Didelphidae	*Didelphis virginiana*	8	[42,43]
Sciuridae	*Tamias striatus*	7	[38]
Suidae	*Sus scrofa*	5	[39,46]
Emydidae	*Terrapene carolina*	5	[38]
Phyllostomidae	*Tonatia bidens*	2	[45]
Cricetidae	*Peromyscus leucopus*	1	[43]
Dasypodidae	*Dasypus novemcintus*	1	[37]
	**Avian**		
Phasianidae	*Gallus domesticus*	4	[39]
Cardinalidae	*Cardinalis cardinalis*	1	[42]
Paridae	*Poecile carolinensis*	1	[42]
Columbidae	*Zenaida macroura*	1	[42]
Tamnophilide	*Taraba major*	1	[45]
Grand total		1647	

Unidentified mammals: Leporidae (*n* = 119); Didelphidae (*n* = 68); Procyonidae (*n* = 58); Sciuridae (*n* = 8); Murid (*n* = 4); ¸); Myomorpha (*n* = 4) = 261. Unidentified birds: Passeriformes (*n* = 10); Columbiformes (*n* = 5); Ciconiiformes (*n* = 1) Quail (*n* = 1) = 17.

**Table 3 insects-12-00967-t003:** Chronological summary of publications on natural infection of *Ae. albopictus* with arboviruses of medical and veterinary importance.

Year	Country	Arbovirus	Detection Technique	Author	Observations
1991	USA	EEEV	IFAA, Viral isolation, PRNT	[48]^H^	EEEV was isolated from 14 pools of females
1991	USA	Keystone virus	IFAA, Viral isolation, PRNT	[48]^H^	Keystone virus was isolated from a pool of females
1993	Brazil	DENV-1	Viral isolation, PCR	[58]^V^	DENV-1 was isolated from two pools of 30 larvae.
1995	USA	CVV	Viral isolation, IFAA	[49]^H^	CVV was isolated from a pool of ~100 females
1995	Mexico	DENV-2 and DENV-3.	Viral isolation, IFAA, RT-PCR	[61]^V^	DENV-2 and DENV-3 were isolated from a pool of ten males
1999	USA	La Crosse Virus	Viral isolation, RT-PCR	[62]^V^	Larvae reared to adults. La Crosse virus was isolated from two pools of 58 females.
1999	Brazil	DENV-3	PCR	[63]^V^	DENV-3 was isolated from three pools of 30 larvae.
2000	USA	WNV	RT-PCR	[50]^H^	WNV was isolated from a pool of two females
2002	Colombia	DENV-1 and DENV-2	RT-PCR	[11]^H^	Two pools of 26 females were positive for DENV-1, and DENV-2, respectively.
2003	Brazil	DENV-2	RT-PCR	[64]^V^	DENV-2 was identified from 33 pools of 1650 larvae. Two Pools of 100 larvae were coinfected with DENV-1 and DENV-2.
2003–2005	USA	WNV	Viral isolation, ELISA	[46]^H^	WNV was isolated of females
2010	USA	CVV	Viral isolation, RT-PCR	[51]^H^	CVV was isolated from three pools of *Ae. albopictus*
2011	Brazil	DENV-2 and DENV-3	RT-PCR	[65]^V^	DENV-2 was found in three individual larvae. DENV-2 was found in a pool of six larvae.An individual larva coinfected with DENV-2 and DENV-3.
2007	Brazil	DENV-2 and DENV-3	Viral isolation, IFAA, RT-PCR	[66]^V^	Larvae reared to adults. DENV-3 was isolated from a pool of 35 adults. One pool of 41 adults coinfected with DENV-2 and DENV-3
2010	Mexico	Dengue, serotype unknown	RT-PCR	[67]^V^	Larvae reared to adults. DENV was found from a pool of four females.
2014–2015	Brazil	DENV-3	RT-PCR	[68]^V^	Two pools of 20 males were positive for DENV-3
2015	Costa Rica	DENV-1, DENV-2, DENV-4	RT-PCR	[55]^H^	Three pools of 60 females were positive for DENV-1, DENV-2, and DENV-4, respectively.
2015	Costa Rica	Dengue, serotype unknown	RT-PCR	[55]^V^	One pool of 20 males were positive for DENV
2015	Brazil	ZIKV	RT-qPCR	[59]^V^	One pool of 33 larvae were positive for ZIKV
2016	Colombia	DENV-2	RT-PCR	[52]^H^	One pool of four females were positive for DENV-2
2016	Mexico	ZIKV	RT-qPCR	[14]^H^	One pool of six females were positive for ZIKV
2016	Colombia	DENV-2	RT-PCR	[60]^H^	Twenty pools of females were positive for DENV-2
2017	Colombia	DENV-4	RT-PCR	[60]^V^	Five pools of 31 males were positive for DENV-4
2017	Mexico	ZIKV	RT-qPCR	[53]^H^	Seven pools of 78 females were positive for ZIKV
2017	Brazil	DENV-4 and ZIKV	Viral isolation, and RT-PCR	[57]^V^	Eggs were reared until adulthood. Two pools were positive for DENV-4 and two pools were positive for ZIKV.
2018–2019	Brazil	ZIKV and YFV	RT-PCR	[47]^V^	Eggs were reared until adulthood. One pool of nine females were positive for YFV. One pool of 32 females and another pool of two males were positive for ZIKV.
2019	Brazil	DENV-1 and ZIKV	RT-qPCR	[54]^H^	One pool of 10 females and another pool of 15 females were positive for ZIKV and DENV-1, respectively.

Indirect fluorescent antibody assay (IFAA); Polymerase Chain Reaction (PCR); Quantitative reverse transcription PCR (RT-qPCR); Plaque-reduction neutralization Test (PRNT); Eastern equine encephalitis virus (EEEV); Cache Valley virus (CVV); West Nile virus (WNV); Yellow fever virus (YFV); Publications with super index ^H^ and ^V^ indicates horizontal and vertical transmission, respectively.

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
