# Peer review of "An Updated Review of the Invasive Aedes albopictus in the Americas; Geographical Distribution, Host Feeding Patterns, Arbovirus Infection, and the Potential for Vertical Transmission of Dengue Virus"

_insects, 2021, doi:10.3390/insects12110967_

Round 1

Reviewer 1 Report

Manuscript title: An updated review of the invasive Aedes albopictus in the Americas; geographical distribution, host feeding patterns, arbovirus infection and the potential for vertical transmission of Dengue virus

            The manuscript reflects a good review collection in the area that is of high medical significance. Yet, the language, formatting and contents of simple abstract to conclusion needs reformatting. The contents of abstract shall start with a background, scope and significant methodologies along with appropriate keywords. Introduction shall describe the need for current study along with current status and description of species and importance of work carried out followed by scope. Rather than presenting consolidated results of previous works, a more comprehensive insight shall be described.

The manuscript shall be resubmitted with major corrections carried out to be considered for publication.

Author Response

The manuscript reflects a good review collection in the area that is of high medical significance. Yet, the language, formatting and contents of simple abstract to conclusion needs reformatting. The contents of abstract shall start with a background, scope and significant methodologies along with appropriate keywords. Introduction shall describe the need for current study along with current status and description of species and importance of work carried out followed by scope. Rather than presenting consolidated results of previous works, a more comprehensive insight shall be described.

The manuscript shall be resubmitted with major corrections carried out to be considered for publication.

Reply: We add a brief background in the abstract as you suggest.

We appreciate your words in favor of improving the manuscript, but we also regret not having convinced you of the corrections made in the work. English grammar was checked by a native English language academic. The other reviewers have already approved the manuscript, and we hope that you will also approve it.

Reviewer 2 Report

The authors have answered my concerns.  

Author Response

Thanks for your comments

Reviewer 3 Report

Significant improvements to the manuscript. There remains several scientific names (i.e., genus and species) that are not italicized throughout in body and figures/tables. 

Author Response

Significant improvements to the manuscript. There remains several scientific names (i.e., genus and species) that are not italicized throughout in body and figures/tables. 

REPLY: line 153, Aedes albopictus was written in italics.

            line 159, Aedes albopictus was written in italics.

            line 160, Homo sapiens was written in italics.

            line 161, Canis lupus was written in italics.

            line 161, Rattus norvegicus was written in italics.

            line 161, Felis silvestris was written in italics.

This manuscript is a resubmission of an earlier submission. The following is a list of the peer review reports and author responses from that submission.

Round 1

Reviewer 1 Report

Reviewer comments

Manuscript title: An updated review of the invasive Aedes albopictus in the Americas; the minimum infection rate suggests that is more efficient in the vertical than horizontal transmission of arbo viruses.

The manuscript entitled ‘An updated review of the invasive Aedes albopictus in the Americas; the minimum infection rate suggests that is more efficient in the vertical than horizontal transmission of arbo viruses’ reviews the status of invasive Aedes albopictus in the Americas.

Title should not be a statement – shall be changed to a more concise and meaningful one. Rather concentrating on published papers, abstract shall provide a proper literature insight on what the review wants to convey. Better sets of keywords shall be used. The whole manuscript shall be re checked and altered for language errors. Introduction shall follow a coherent order. Emphasis is given towards the vector or the virus?? A more detailed information on the impact of virus on human health status shall be added..not just their geographical distribution. Statistical analysis methods shall be included in a separate section.. 2.1. or so. Formula shall be included in equation format. Section headings shall be made concise and understandable. Rather than presenting the information from previous publications, the whole manuscript shall be reformatted as an analytical interpretation based on the literature review. The manuscript at this stage only seems to be a coalition of preious results with a little interpretation.

Author Response

  1. a) Title should not be a statement – shall be changed to a more concise and meaningful one

REPLY: The title was modified as "An updated review of the invasive Aedes albopictus in the Americas; geographical distribution, host feeding patterns, arbovirus infection, and the potential for vertical transmission of Dengue virus".

Although we also accept any title that you suggest

  1. b) Rather concentrating on published papers, abstract shall provide a proper literature insight on what the review wants to convey

REPLY: The abstract was modified as suggested

  1. c) Better sets of keywords shall be used

REPLY: The sets of keywords were modified as: Asian tiger mosquito, feeding pattern, minimum infection rate, emerging arboviruses, dengue virus.

  1. d) The whole manuscript shall be re checked and altered for language errors.

REPLY: English grammar was checked by a native English speaker.

  1. e) Introduction shall follow a coherent order. Emphasis is given towards the vector or the virus?? A more detailed information on the impact of virus on human health status shall be added..not just their geographical distribution.

REPLY: In the introduction, we add the burden of dengue in America because it is the disease that causes the highest morbidity and mortality (line 58 to 65). This virus is transmitted by Ae. albopictus and could impact transmission dynamics in the region.

  1. f) Statistical analysis methods shall be included in a separate section.. 2.1. or so.

REPLY: a statistical analysis subsection was established.

  1. g) Formula shall be included in equation format.

REPLY: The minimum infection rate was written in the formula format.

  1. h) Section headings shall be made concise and understandable.

REPLY: We consider that the title of section 6 should be modified. “The minimum infection rate estimated in vertical and horizontal transmission of arboviruses” was changed to ” The minimum infection rate of the dengue virus”.

  1. i) Rather than presenting the information from previous publications, the whole manuscript shall be reformatted as an analytical interpretation based on the literature review. The manuscript at this stage only seems to be a coalition of previous results with a little interpretation.

REPLY: We have a disagreement with this point. However, we recognize that some points need clarification to improve the work. We hope that with all the corrections made we can satisfy the quality of the publication.

Reviewer 2 Report

The stated aim of the manuscript “An updated review of the invasive Aedes albopictus in the Americas; the minimum infection rate suggests that is more efficient in the vertical than horizontal transmission of arboviruses” was to review the literature to update the reported distribution range, blood feeding patterns, and compare published minimum infection rates relative to vertical and horizontal transmission of various arboviruses. The authors conclusions that Ae. albopictus is an invasive mosquito that readily adapts to new areas, is highly competent to transmit several arboviruses via transovarial transmission, may contribute to endemic transmission as a bridge vector for emerging arboviruses between sylvan, rural, and urban areas are interesting, but not necessarily novel.

Generally the manuscript is well-written and contributes updated valuable information towards better understanding geographic distribution of Aedes albopictus, and its capacity/influence on arboviral maintenance and transmission.

Specifically, there are several cases throughout the manuscript in which sentence structure and choice of wording needs revision for accuracy and clarity (e.g., Lines 27, 28, 55, 71, 72, 184, 199, 201, 266). Additionally there are style inconsistencies and contradictions, particularly the spelling-out of numbers less than or equal to ten, as well as introducing abbreviations upon first mention but continuing to spell-out and use abbreviations sporadically throughout. The abbreviation “EE.UU.” occurs five times in the body of the paper, and eight times in Tables 1 & 3 – what does is it an abbreviation for? Correct abbreviation for Venezuelan equine encephalitis virus in Line 164. Finally, scientific names should be italicized (Lines 143, 149-151).

Author Response

Reviewer 2

  1. a) The stated aim of the manuscript “An updated review of the invasive Aedes albopictus in the Americas; the minimum infection rate suggests that is more efficient in the vertical than horizontal transmission of arboviruses” was to review the literature to update the reported distribution range, blood feeding patterns, and compare published minimum infection rates relative to vertical and horizontal transmission of various arboviruses. The authors conclusions that albopictus is an invasive mosquito that readily adapts to new areas, is highly competent to transmit several arboviruses via transovarial transmission, may contribute to endemic transmission as a bridge vector for emerging arboviruses between sylvan, rural, and urban areas are interesting, but not necessarily novel.

REPLY: We agree that there are many studies on the distribution of Ae. albopictus and its role as a vector of arboviruses. However, to our knowledge, there is no work comparing the minimum infection rate between vertical and horizontal transmission studies. We believe that this is an important contribution.

  1. b) Generally the manuscript is well-written and contributes updated valuable information towards better understanding geographic distribution of Aedes albopictus, and its capacity/influence on arboviral maintenance and transmission.

REPLY: Thanks for the words, our intention is to decipher the role of Ae. albopictus as a potential vector of arbovirus in America, mainly in the transmission and maintenance dynamics of the dengue virus in inter-epidemic periods.

  1. c) Specifically, there are several cases throughout the manuscript in which sentence structure and choice of wording needs revision for accuracy and clarity (e.g., Lines 27, 28, 55, 71, 72, 184, 199, 201, 266).

REPLY: The mentioned sentences were revised and corrected as suggested.

Lines 27-28 were part of the summary, and it was rewritten.

Line 55 was deleted and rewritten.

The introduction was slightly modified in order and the original line 71 and 72 was deleted.

Line 184 was revised and rewritten.

Line 199 was deleted.

Line 201 was revised and rewritten.

Line 266 was revised and rewritten.

  1. d) Additionally there are style inconsistencies and contradictions, particularly the spelling-out of numbers less than or equal to ten, as well as introducing abbreviations upon first mention but continuing to spell-out and use abbreviations sporadically throughout.

REPLY: Thanks for the observation, the inconsistencies were reviewed and corrected as suggested.

  1. e) The abbreviation “EE.UU.” occurs five times in the body of the paper, and eight times in Tables 1 & 3 – what does is it an abbreviation for?

REPLY: We change “EE.UU.” to “USA”.

  1. f) Correct abbreviation for Venezuelan equine encephalitis virus in Line 164.

REPLY: The name of the virus was reviewed and corrected as suggested

  1. g) Finally, scientific names should be italicized (Lines 159, 160, 161).

REPLY: The scientific names were italicized.

Reviewer 3 Report

This is a well-researched and thought-out article that should be useful to readers of the journal. Below are some relatively minor comments that the authors should consider to improve the paper:

Title – it seems the MIR were calculated for DENV not arboviruses in general. The title could be shortened to A review of invasive Aedes albopictus in the Americas.

Line 71: It was first described in Tennessee (line 117) so this reference to ‘colonized’ and 36 years is confusing.

Readers might be more familiar with USA than EE.UU.

It would be useful for readers to know which viruses were targeted by the PCR studies, perhaps in the Table. Presumably the PCR studies were not designed to detect all arboviruses – they were targeted at certain viruses.  

Line 195 – this paragraph seems to indicate that chikungunya is not transmitted by Ae. albopictus in the Americas.

Line 205 – to be a competent vector there need to have been transmission studies. The references indicate only that these viruses have been found in the Ae. albopictus.

Line 207 – if these publications only searched for DENV and ZIKA this does not preclude there being other viruses present. This should be made clear.  Also, which papers are these – consider citing Table 3 as a source for the references as in line 228.

Line 210 – which papers were these? Once again consider citing the Table 3 which seems to provide the references.

Line 229 – these publications are not stipulated – are they in Table 3? They seem to all be DENV? If they were then it should be stated the MIR was calculated for DENV.

Line 232 – were there any differences in the pool sizes?  For MIR it is assumed that there is only one infected individual in each pool so differences in pool size might make a difference.

Line 240 - the authors should provide possible reasons for horizontal transmission being more common with Ae aegypti

Author Response

Reviewer 3

This is a well-researched and thought-out article that should be useful to readers of the journal. Below are some relatively minor comments that the authors should consider to improve the paper:

  1. a) Title – it seems the MIR were calculated for DENV not arboviruses in general. The title could be shortened to A review of invasive Aedes albopictus in the Americas.

REPLY: In the first version, the MIR was calculated only for arboviruses in general. Now we also estimate the MIR for dengue. Interpretation was included in the manuscript.

  1. b) Line 71: It was first described in Tennessee (line 117) so this reference to ‘colonized’ and 36 years is confusing.

REPLY: The paragraph was rewritten and cited as suggested.

  1. c) Readers might be more familiar with USA than EE.UU.

REPLY: We change “EE.UU.” to “USA” as you suggest.

  1. d) It would be useful for readers to know which viruses were targeted by the PCR studies, perhaps in the Table. Presumably the PCR studies were not designed to detect all arboviruses – they were targeted at certain viruses.

REPLY: In Table 3, there is a column that indicates the technique used to identify arboviruses.

  1. e) Line 195 – this paragraph seems to indicate that chikungunya is not transmitted by albopictus in the Americas.

REPLY: It is correct, currently in America the chikungunya virus has not been identified in field-caught Aedes albopictus.

  1. f) Line 205 – to be a competent vector there need to have been transmission studies. The references indicate only that these viruses have been found in the Ae. albopictus.

REPLY: The paragraph was corrected, the word "competent vector" was removed.

  1. g) Line 207 – if these publications only searched for DENV and ZIKA this does not preclude there being other viruses present. This should be made clear.  Also, which papers are these – consider citing Table 3 as a source for the references as in line 228.

REPLY: The paragraph on line 207 and 228 were cited as suggested.

  1. h) Line 210 – which papers were these? Once again consider citing the Table 3 which seems to provide the references.

REPLY: I think he got the wrong paragraph, in the previous version line 210 is empty. However, we have cited another paragraphs that had no reference.

  1. i) Line 229 – these publications are not stipulated – are they in Table 3? They seem to all be DENV? If they were then it should be stated the MIR was calculated for DENV.

REPLY: In the current version, the MIR was estimated for dengue and the interpretation and discussion was included.

  1. j) Line 232 – were there any differences in the pool sizes?  For MIR it is assumed that there is only one infected individual in each pool so differences in pool size might make a difference.

REPLY: Yes, that's correct, the pool size in each study ranged from 10 to 50 mosquitoes. In the present work, the comparison of the MIR was an approximation. However, it provides the probability of finding arboviruses per -1000 Ae. albopictus tested.

  1. K) Line 240 - the authors should provide possible reasons for horizontal transmission being more common with Ae aegypti.

REPLY: Some reasons for the effectiveness of Aedes aegypti in horizontal transmission was included as suggested (line 260-265).